# A Review of Endobronchial-Ultrasound-Guided Transbronchial Intranodal Forceps Biopsy and Cryobiopsy

**DOI:** 10.3390/diagnostics14090965

**Published:** 2024-05-06

**Authors:** Michel Chalhoub, Bino Joseph, Sudeep Acharya

**Affiliations:** Staten Island University Hospital, 475 Seaview Avenue, Staten Island, NY 10305, USA; bjoseph4@northwell.edu (B.J.); sacharya1@northwell.edu (S.A.)

**Keywords:** EBUS, EBUS-guided transbronchial needle aspirate, EBUS-guided intranodal forceps biopsy, EBUS-guided cryobiopsy, advances in interventional pulmonology

## Abstract

Benign and malignant mediastinal lesions are not infrequently encountered in clinical practice. Mediastinoscopy has long been considered the gold standard in evaluating mediastinal pathology. Since its introduction into clinical practice, endobronchial-ultrasonography-guided transbronchial fine needle aspiration (EBUS-TBNA) has replaced mediastinoscopy as the initial procedure of choice to evaluate mediastinal lesions and to stage lung cancer. Its diagnostic yield in benign mediastinal lesions and less common malignancies, however, has remained limited. This has led different proceduralists to investigate additional procedures to improve the diagnostic yield of EBUS-TBNA. In recent years, different published reports concluded that the addition of EBUS-guided intranodal forceps biopsy (IFB) and transbronchial cryobiopsy (TBCB) to EBUS-TBNA increases the diagnostic yield especially in benign mediastinal lesions and uncommon mediastinal malignancies. The purpose of this review is to describe how EBUS-IFB and EBUS-TBCB are performed, to compare their diagnostic yields, and to discuss their limitations and their potential complications. In addition, the review will conclude with a proposed algorithm on how to incorporate EBUS-IFB and EBUS-TBCB into clinical practice.

## 1. Introduction

Lung cancer is the second most common cancer in men and women and is the leading cause of cancer death in both genders [1,2]. One of the most common sites of the metastatic spread of lung cancer is the mediastinal lymph nodes (LNs) [3,4]. Prior to initiating any therapy for lung cancer, it is crucial to perform accurate staging [5]. This usually entails adequate sampling of the mediastinal LNs [6]. In clinical practice, both benign as well as malignant mediastinal lesions are encountered [7]. In most instances, radiological and clinical features are usually not enough to establish a definitive diagnosis and a treatment plan [8]. Endobronchial-ultrasound-guided transbronchial fine needle aspiration (EBUS-TBNA) emerged as an excellent, minimally invasive tool, to sample mediastinal lesions on an outpatient basis [9,10]. It has been shown in multiple studies to be at least as accurate as mediastinoscopy with less potential complications [11,12]. One drawback of EBUS-TBNA remains the amount of tissue that is removed with each needle pass [13]. Even though the yield in carcinoma is excellent, the yield of EBUS-TBNA in benign conditions and lymphoma and some other malignancies remains less than optimal [14,15]. In a world where immunotherapy and targeted therapy has revolutionized the way lung cancer is treated, biopsy techniques that ensure adequate tissue for proper and adequate genetic sequencing are mandatory.

In recent years, transbronchial lung cryobiopsy has emerged as an alternative technique for surgical lung biopsy (SLB) in diffuse parenchymal lung diseases (DPLD) [16,17,18,19]. The advantage of this technique is it offers a less invasive means of obtaining lung tissue in DPLD compared to SLB and without the crush artifact that is often seen with transbronchial biopsies obtained by forceps biopsies [16]. Different studies have shown a very good correlation of transbronchial lung biopsies with SLB [17,18]. In its most recent practice guidelines of idiopathic pulmonary fibrosis, the authors have recommended the use of transbronchial lung cryobiopsy over SLB in experienced centers [18,19]. 

This has led different investigators to study different techniques using EBUS to guide biopsy forceps and cryo-probes to perform biopsies of mediastinal lesions and lymph nodes in the hope of acquiring more adequate tissue for all needed pathologic and cytologic evaluations [20,21]. This review will summarize the utility of EBUS-guided intranodal forceps biopsies (EBUS-IFB) and transbronchial cryonodal biopsies (EBUS-TBCB) and compare those techniques to EBUS-TBNA. The limitations as well as potential complications of these procedures will also be described. 

## 2. EBUS-Guided Intranodal Forceps Biopsy Technique

EBUS-IFB basically uses linear EBUS to guide mini forceps into the desired lymph node or lesion to obtain a tissue biopsy. It is suggested as an additional procedure during EBUS-TBNA and not as a stand-alone procedure [22]. Multiple studies have shown that adding EBUS-IFB to EBUS-TBNA increases the diagnostic yield significantly. After performing the EBUS-guided needle aspirations, mini forceps are passed and introduced into the lymph node using the same puncture site created by the aspiration needle tract. Vascular supply of the lymph node or the structure being biopsied is then evaluated and, under EBUS guidance, forceps biopsies are then obtained. When using a 19-gauge needle to perform the initial aspiration, the puncture site is usually more readily identifiable to introduce the mini forceps [23]. When using a 21- or 22-gauge needle, however, the puncture site might not be as readily identifiable and the operator must maintain the scope in the same position used to perform the TBNA to facilitate the forceps’ entry though the puncture site. This is accomplished by keeping the EBUS image of the lymph node at the same level throughout the forceps’ introduction. Using some lymph node characteristics and other anatomic landmarks to keep the EBUS scope at the same position has been advocated by some authors [22,23]. An electrocautery knife (Olympus KD-31C-1; Erbe VIO 300D-400W) has been used by some to form a tract that allows 1.9 mm biopsy forceps to be introduced into the desired lymph node and to obtain tissue biopsies [24]. That can theoretically ensure more tissue retrieval compared to using mini biopsy forceps, but no comparative studies are available to date. 

## 3. EBUS-Guided Intranodal Forceps Biopsy Results

When performing EBUS-IFB in addition to EBUS-TBNA, the overall reported diagnostic yield is usually improved [24]. When considering different etiologic conditions, the addition of EBUS-IFB to EBUS-TBNA offers the most benefit in non-malignant mediastinal lesions in comparison to malignant lesions [25]. In a report by Agarwal et al., the overall diagnostic yield when EBUS-IFB was combined with EBUS-TBNA was 92% compared to 67% when EBUS-TBNA was performed alone [26]. Chrissian et al. compared EBUS-TBNA to EBUS-IFB in mediastinal LNs and found an overall diagnostic yield of 81% and 91%, respectively (*p* value of 0.009). When combining EBUS-IFB with EBUS-TBNA, the overall diagnostic yield improved to 97% with a *p* value of less than 0.001 when compared to EBUS-TBNA alone [22].

In another report, Bramley et al. reported the results of EBUS-guided cautery-assisted IFB compared to EBUS-TBNA. The report included 50 patients with 111 lymph nodes biopsied. In this series, the authors used larger forceps than usually used in other reports of EBUS-IFB, and the authors reported their results in per-nodal analysis rather than per-patient analysis. The results were consistent with other series where EBUS-TBNA performed better in malignant lymph nodes than in non-malignant lymph nodes. The diagnostic yield was 100% in malignant lymph nodes compared to 78% for non-malignant lymph nodes with a *p* value of 0.001 [27]. It is worth noting that in this report the authors included lymphoma in malignant lymph nodes whereas in other reports, lymphoma was reported separately. EBUS-IFB was able to identify granulomatous inflammation in 17 out 19 lymph node biopsies with a diagnostic yield of 89%, whereas EBUS-TBNA alone was able to identify granulomatous inflammation in 6 out of 19 lymph nodules biopsied with a diagnostic yield of only 32%. There were only two lymph nodes secondary to infection and EBUS-IFB was able to diagnose both (100% yield), whereas EBUS-TBNA was able to identify one out of the two lymph nodes (50% yield) [27].

When evaluating the diagnostic yield in lymphoma, combining EBUS-IFB with EBUS-TBNA increased the overall diagnostic yield from 30% when EBUS-TBNA was performed alone to 86% when both modalities were combined. In the same review, Agarwal et.al found an increased diagnostic yield when EBUS-IFB was combined with EBUS-TBNA compared to EBUS-TBNA alone in sarcoidosis. The yield increased from 58% when only EBUS-TBNA was performed to 93% when EBUS-IFB was combined with EBUS-TBNA (*p* value of 0.00001) [28]. The following table [Table 1] summarizes the diagnostic yield of EBUS-IFB in comparison to EBUS-TBNA alone.

## 4. EBUS-Guided Intranodal Forceps Biopsy Related Complications

EBUS-IFB is generally considered a safe procedure with a low risk of major complications. In addition to the general complications related to anesthesia, procedure-specific complications are rare. The EBUS-IFB-related specific complications include bleeding 0.8%, pneumothorax 1%, pneumomediastinum 1%, and potentially, mediastinal infections. These EBUS-IFB-related complications are slightly higher than those reported when only EBUS-TBNA is performed. A pneumothorax incidence of 0.03% and bleeding incidence of 0.68% are reported with EBUS-TBNA alone. In published reports, the bleeding site was controlled successfully by simply injecting cold saline. Most cases of pneumomediastinum and pneumothoraxes were asymptomatic and required no specific therapy. All resolved with watchful waiting. Mediastinal infection, a potential theoretical complication of IFB and TBCB, has not been reported in published series [23,26,27]. 

## 5. EBUS-Guided Transbronchial Cryobiopsy 

The procedure for obtaining EBUS-guided transbronchial cryo-biopsy EBUS-TBCB is very similar to EBUS-IFB. The procedure is performed using anesthesia with either a laryngeal mask or an endotracheal tube [28]. After performing the TBNA, a 1.1 mm cryoprobe is introduced into the working channel of the EBUS scope. The probe is then carefully pushed into the site of the previous TBNA site [29]. Some authors used a high-frequency knife to make an opening to facilitate the probe entry into the lymph node or the lesion being biopsied [30]. Using the EBUS guidance, the desired position of the cryoprobe is confirmed. Once the position is confirmed, the cryoprobe is activated for 3 to 7 s and the probe, along with the scope, is removed “en-block”. The tissue is thawed in saline and fixed with formalin. The entry site is then inspected for any significant bleeding prior to attempting another biopsy [31] [Figure 1]. 

## 6. EBUS-Guided Transbronchial Cryobiopsy Results

When compared to EBUS-TBNA alone, EBUS-TBCB performed better in obtaining a diagnosis in all cases of mediastinal lesions and adenopathy. Zhang and Guo evaluated the diagnostic yield when EBUS-TBCB was added to EBUS-TBNA in 197 patients. The overall diagnostic yield of EBUS-TBNA was about 80% compared to 92% for EBUS-TBCB. The diagnostic yield increased to 93% when both modalities were combined, with a *p* value of 0.001. In the same report, EBUS-TBCB performed significantly better in less common mediastinal malignancies when compared to EBUS-TBNA; less common mediastinal malignancies included carcinoid tumors, sarcomatoid lung cancer, lymphoma, seminoma, and thymic carcinoma. In the more common lung cancer subtypes, however, there was no significant difference between either modality: 94.1% and 95.6% for EBUS-TBNA and EBUS-TBCB, respectively. Common lung cancer types in that series included adenocarcinoma, squamous cell, large cell, small cell, and non-small cell lung cancer not otherwise specified [31].

In another open label randomized trial performed in Germany and China, Fan and Zhang compared the combination of EBUS-TBCB and EBUS-TBNA to EBUS-TBNA alone in mediastinal diseases. A total of 135 subjects underwent EBUS-TBNA alone and were compared to 136 subjects who underwent EBUS-TBNA combined with EBUS-TBCB. The overall diagnostic yield increased from 81% when EBUS-TBNA was performed alone to 93% when EBUS-TBCB was combined with EBUS-TBNA (*p* = 0.0039) [8]. In subgroup analysis, the diagnostic yield was similar in malignant mediastinal neoplasms with a diagnostic yield of 94% (81 out of 86 subjects) for combined EBUS-TBCB and EBUS-TBNA versus 91% (81 of 89 subjects) for EBUS-TBNA alone. Mediastinal malignancies included primary lung cancer, metastatic disease, and uncommon mediastinal malignancies. In benign mediastinal lesions, however, the diagnostic yield for combined EBUS-TBCB and EBUS-TBNA was significantly higher than EBUS-TBNA alone: 94% (45 of 48 subjects) compared to 67% (32 of 48 subjects), respectively. Benign mediastinal disorders in that series included pneumoconiosis, sarcoidosis, and tuberculosis [7]. These findings are in line with the previous report by Zhang and Guo. In contradiction, however, this series did not show a statistically significant difference in the diagnostic yield in non-common lung cancer between the combined procedures compared to EBUS-TBNA alone even though there was a tendency for improved diagnostic accuracy with the combined procedure. The overall diagnostic yield for the combined procedure was 76% (13 out of 17 subjects) compared to 59% (10 out of 17 subjects) with a *p* value of 0.47 for EBUS-TBNA alone [8,31]. A firm conclusion on whether combined EBUS-TBCB and EBUS-TBNA offers advantage over EBUS-TBNA alone in non-common lung cancer remains difficult to establish based on the small number of subjects included in the subgroup analysis in both series. As more studies and case series become available, this issue might be better clarified. In the report by Zhang and Guo, the combination of EBUS-TBCB and EBUS-TBNA was superior for the diagnosis of lymphoma compared to EBUS-TBNA alone. The diagnostic yield of the combined procedure was 87.5% compared to 12.5% when EBUS-TBNA alone was performed [27]. In addition, all cases of lymphoma were able to be subclassified when EBUS-TBCB was combined with EBUS-TBNA compared to EBUS-TBNA alone [8,21]. 

In the same report, the combined procedure of EBUS-TBCB and EBUS-TBNA assured genetic sequencing in 93.3% of patients compared to 73.5% when EBUS-TBNA was performed alone [31]. These findings were similar to the findings of the study by Fan and Zhang where genomic testing and PDL-1 immunohistochemistry assays were adequate in 97% of cases (37 out of 38 subjects) compared to 79% (30 out 37 subjects) for EBUS-TBNA alone (*p* value of 0.033) [8].

When performing EBUS-TBCB in addition to EBUS-TBNA, the procedure is more time consuming. The additional time needed is reported to be about 2 min [31]. In the report by Jin Zhang, performing EBUS examination was accomplished in 31.9 + 9.1 min [27]. Adding TBCB required an additional 11.7 + 5.3 min, whereas adding TBNA required only an additional 9.4 + 2.6 min (*p* < 0.001) [31]. In general, the majority of time spent during an EBUS-TBNA is for handling the specimen acquired, while the extra time spent during an EBUS-TBCB is mostly due to the need to remove the scope and reidentification of the biopsy site and introducing the cryoprobe [31].

The following table [Table 2] summarizes the diagnostic yield of EBUS-TBCB in comparison to EBUS-TBNA alone. 

## 7. EBUS-Guided Transbronchial Cryobiopsy Related Complications

The potential complications with EBUS-TBCB are very similar to the complications encountered with EBUS-IFB or EBUS-TBNA. The procedure is considered safe, and it is thought that the morbidities and mortality associated with this procedure remain significantly less than those seen with mediastinoscopy or VATS [28]. Commonly described complications include pneumothorax, seen in about 1%; pneumomediastinum, seen in about 1%; and bleeding, seen in about 10–14% [32]. In all reported series, pneumomediastinum and pneumothoraxes did not require any intervention. They resolved spontaneously. Most cases of bleeding were mild and did not require any specific therapy. Bleeding resolved spontaneously and with instillation of cold saline and on occasions epinephrine [33]. Other reported complications included cough (14%), dyspnea (1%), and hemoptysis (2%) [8,31]. In most of the published series, EBUS-TBCB was combined with EBUS-TBNA in the same patients. It is difficult therefore to attribute the complications to TBCB since TBNA was performed in the same settings.

The following table [Table 3] summarizes the complications of EBUS-TBCB and compares them to EBUS-IFB. 

## 8. EBUS-Guided Transbronchial Cryobiopsy Compared to EBUS-Guided Intranodal Forceps Biopsy

Until recently, there were no comparative studies that compared the diagnostic yield of adding either EBUS-TBCB or EBUS-IFB to EBUS-TBNA. In a recent review, Cheng and colleagues published a randomized study that compared the additional benefit of adding EBUS-TBCB compared to adding EBUS-IFB to EBUS-TBNA [32]. In that report, the authors enrolled a total of 155 subjects. In 77 subjects, they performed four EBUS-TBNAs followed by three EBUS-IFBs followed by one EBUS-TBCB; and in 78 subjects, they performed four EBUS-TBNAs followed by one EBUS-TBCB followed by three EBUS-IFBs.

The overall diagnostic yield of EBUS-TBNA and EBUS-IFB was similar to that of EBUS-TBNA and EBUS-TBCB [32]. For non-small lung cancer, however, genetic testing was statistically superior when comparing the addition of EBUS-TBCB to EBUS-TBNA to the addition of EBUS-IFB to EBUS-TBNA: 100% compared to 89.5%, respectively, with a *p* value of 0.036. In benign mediastinal conditions when EBUS-TBCB was added to EBUS-TBNA, the diagnostic yield increased from 59.6% to 78.7% with a *p* value of 0.044, whereas when EBUS-IFB was added to EBUS-TBNA in benign mediastinal conditions, the diagnostic yield was not statistically different: 59.6% compared to 66% with a *p* value of 0.522. Benign mediastinal conditions included Castleman’s disease, sarcoidosis, tuberculosis, tumor-related granulomas, and pneumoconiosis (38). In the same review, when EBUS-IFB alone and EBUS-TBCB alone were compared, EBUS-TBCB performed statistically better with a diagnostic yield of 85.7% compared to 70.8% for EBUS-IFB (*p* value of 0.001). The subgroup analysis showed that EBUS-TBCB performed statistically better compared to EBUS-IFB in common lung cancer: 92.6% compared to 77.8% with a *p* value of 0.008. Finally, in uncommon tumors and benign disorders, even though the diagnostic yield of EBUS-TBCB was higher than EBUS-IFB, this result did not reach statistical significance [32].

## 9. Rare Mediastinal Pathologies Diagnosed with EBUS-TBCB

Some case reports of Yuki and colleagues reported a case of esophageal submucosal leiomyoma diagnosed with EBUS-TBCB that could not diagnosed with EBUS-TBNA alone, and where EBUS-IFB was unsuccessful [34]. In that report, the EBUS-FNA specimen showed some spindle-shaped cells, but the tissue obtained was not enough for definitive diagnosis. EBUS-IFB was attempted, but the forceps could not penetrate the tumor outer capsule. EBUS-TBCB was then performed with no difficulty and a definitive diagnosis of esophageal leiomyoma was established [34]. There were no significant complications reported during that procedure. In a similar fashion, Chichiro et al. reported a case of a thoracic SMARC4-deficient undifferentiated tumor diagnosed with EBUS-TBCB where EBUS-TBNA and EBUS-IFB were unsuccessful [35]. Jing Zhang and Colleagues reported the successful diagnosis of mediastinal seminoma by EBUS-TBCB where EBUS-TBNA was unsuccessful in establishing a definitive diagnosis [36]. Jing Zhang also reported a case of primary mediastinal large B-cell lymphoma that was diagnosed with EBUS-TBCB [37]. These reports suggest that when EBUS-TBNA is unsuccessful in establishing a firm diagnosis, adding EBUS-TBCB might be of value. 

## 10. Proposed Algorithm

Since EBUS-IFB and EBUS-TBCB are relatively new procedures, they are not included in published guidelines and there are no solid recommendations on when to use them. It seems, from reviewing the available data, that EBUS-IFB or TBCB should not be performed as stand-alone procedures at this time. They should be combined with EBUS-TBNA in all cases until further data become available. Since there is no head-to-head comparison between TBCB and IFB, one cannot recommend one procedure over the other. 

The following simplified guidelines could be used [Figure 2]:In a case where lung cancer is diagnosed and genetic testing is available, whether from an extra thoracic site or a lung mass, and mediastinal lymph node staging is required, EBUS-TBNA alone should be performed;In a case with mediastinal lymphadenopathy suspected from a malignancy other than lung cancer, EBUS-TBNA can be performed alone;In a case with common or non-common mediastinal malignancy, combined EBUS-TBNA with either EBUS-TBCB or -IFB should be performed targeting the lesion that will ensure the highest staging. A minimum of four passes in total is suggested: two TBNA and two TBCB or IFB are thought to be adequate. Performing combined procedures for the rest of the lymph nodes in the same patients will likely prolong the procedure time significantly without adding a significant increase in the diagnostic yield;In a case where lymphoma is suspected, combined EBUS-TBNA and either EBUS-TBCB or -IFB is suggested over EBUS-TBNA alone;In a case where benign mediastinal pathology is suspected, combined EBUS-TBNA and either EBUS-TBCB or IFB is suggested over EBUS-TBNA alone. Benign mediastinal pathology commonly diagnosed with the combined procedure includes sarcoidosis, tuberculosis, and pneumoconiosis.

## 11. Conclusions

Guidelines recommend the use of EBUS-TBNA as the initial procedure of choice for mediastinal nodal staging in lung cancer. Its utility, however, can be limited on occasions by its diagnostic yield in certain uncommon lung cancers as well as in genomic testing. Even though the diagnostic yield of EBUS-TBNA in recurrent lymphoma is considered reasonable, its diagnostic yield of de novo cases and the proper histological typing of lymphoma are considered less than optimum. In addition, the diagnostic yield of EBUS-TBNA in benign mediastinal pathology remains limited. The addition of intranodal forceps biopsy or cryobiopsy to TBNA offers a higher diagnostic yield in both malignant and non-non-malignant mediastinal lesions. The procedures can be added to EBUS-TBNA for less than 10 min in general and can be performed with low complication rates.

## Figures and Tables

**Figure 1 diagnostics-14-00965-f001:**
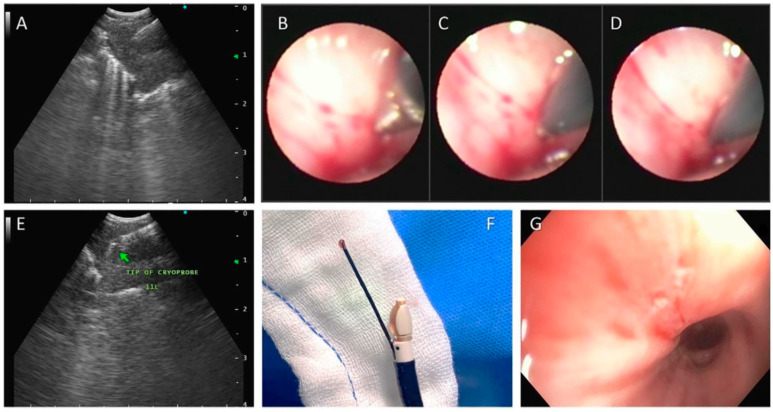
(**A**) Performing endobronchial ultrasound-guided transbronchial needle aspiration (EBUS-TBNA)-EBUS image showing a 19-G needle in station 11L node; steps of inserting 1.1 mm cryo-probe through the puncture site made by TBNA needle, (**B**) tip of cryo-probe at the puncture site, (**C**) pushing the probe, and (**D**) a tip of cryo-probe completely inside the node. (**E**) EBUS image showing the tip of 1.1 mm cryo-probe within the lymph node. (**F**) Olympus EBUS scope (BF-UC 180F) with 1.1 mm cryo-probe in the working channel. The tip of the probe has the lymph node tissue obtained by cryo-nodal biopsy. (**G**) Bronchoscopic view of the puncture site after taking cryo-nodal biopsy. Reproduced with permission [29].

**Figure 2 diagnostics-14-00965-f002:**
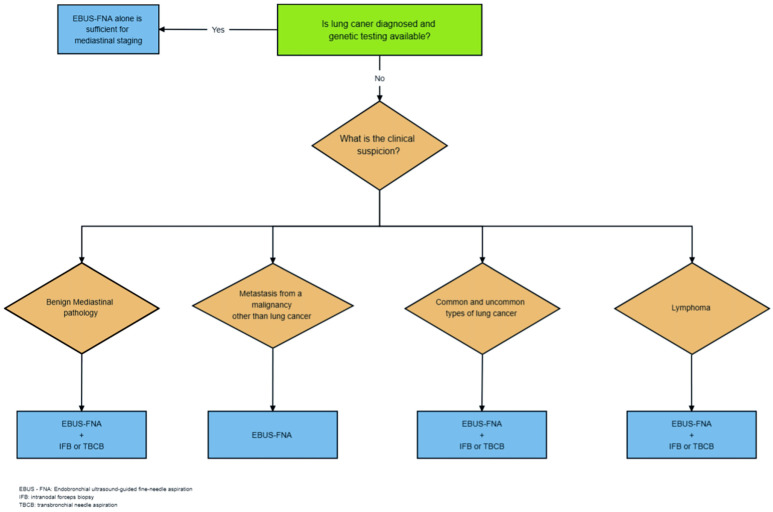
Diagram of the proposed algorithm.

**Table 1 diagnostics-14-00965-t001:** Diagnostic yield in different mediastinal pathologies for combined EBUS-IFB combined with EBUS-TBNA compared to EBUS-TBNA alone.

Diagnostic Yield (%)	EBUS IFB+TBNA (1)	EBUS TBNA	*p* Value
Overall	92–97	67–81	
Lymphoma	86	30	0.003
Sarcoidosis	93	58	<0.00001

(1) Combined EBUS-guided intranodal forceps biopsy and transbronchial needle aspiration.

**Table 2 diagnostics-14-00965-t002:** Diagnostic yield in different mediastinal pathologies for combined EBUS-TBCB with EBUS-TBNA compared to EBUS-TBNA alone.

Diagnostic Yield	EBUS TBCB+TBNA (1)	EBUS TBNA	*p* Value
Overall	93–94 (%)	80–81 (%)	0.001 and 0.0039
Common lung cancer	94.1 (%)	95.6 (%)	NS
Uncommon mediastinal malignancies	91.7 (%) 76 (%)	25 (%) 59 (%)	0.001 NS
Benign mediastinal lesions	94 (%)	67 (%)	0.0009
Mediastinal nodal metastases	99 (%)	99 (%)	NS
Lymphoma	87.5 (%)	12.5 (%)	NA
Genomic testing	93–97 (%)	73.5–79 (%)	0.001

(1) Combined EBUS-guided transbronchial cryobiopsy and transbronchial needle aspiration. Common lung cancers include adenocarcinoma, squamous cell, large cell, small cell, and non-small cell lung cancer not otherwise specified. Uncommon mediastinal malignancies include carcinoid tumors, sarcomatoid lung tumors, lymphoma, seminomas, and thymic carcinomas. Benign mediastinal lesions include sarcoidosis, tuberculosis, and pneumoconiosis. Mediastinal nodal metastases included metastasis from esophageal, pancreatic, and breast cancers. NS = not-significant. NA = not available.

**Table 3 diagnostics-14-00965-t003:** Reported complications for EBUS-IFB and EBUS-TBCB.

Complications (%) *	EBUS-IFB	EBUS-TBCB
Pneumothorax	2.9	1
Pneumomediastinum	2.9	1
Bleeding	6	9
Respiratory failure	2	-
Hemoptysis	-	2

Complications (%) *: Highest reported incidence. EBUS-IFB = EBUS-guided intranodal forceps biopsy. EBUS-TBCB = EBUS-guided transbronchial cryobiopsy. There was one case of pneumonia that occurred 10 days after EBUS-IFB. There was one case of death reported after EBUS-IFB that was attributed to severe aortic stenosis.

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
