# Peer review of "A Review of Endobronchial-Ultrasound-Guided Transbronchial Intranodal Forceps Biopsy and Cryobiopsy"

_diagnostics, 2024, doi:10.3390/diagnostics14090965_

Round 1

Reviewer 1 Report

Comments and Suggestions for Authors

1. The article is a narrative review comparing two minimally invasive thoracic surgical techniques for the diagnosis of mediastinal lesions: EBUS-guided transbronchial cryobiopsy and intrranodal forceps biopsy. The review can be considered satisfactory as an educational article, but not as a research article.

2. The article highlights the features of the two techniques, evaluating their pros and cons. These features are probably well known to thoracic surgery specialists, but may not be for specialists in other techniques who may also be involved in the diagnosis of mediastinal lesions (e.g., interventional radiologists).

3. The review is very readable and clear, but it does not revolutionize the scientific knowledge on the topic. For this reason, I had suggested a low priority for publication.

4. The authors propose an algorithm for the implementation of the two techniques. However, this algorithm appears rather risky and represents a risk for the journal that publishes it. It is not supported by data on case studies treated with it, nor do I have any information on the authors' authority in this field of study. I have extensively emphasized this point in both the response to the authors and to the editor. In my opinion, publishing it without specific consistency of the case studies is too risky for the journal. Furthermore, no specific studies or case-control studies, etc. have been indicated to support it.

5.  Corresponding author: It is not clear who the corresponding author is. Please clearly indicate in the article who readers should contact with any questions or concerns. Furthermore, it would be helpful to provide more details on the authors' affiliation, including their institution and department.

6. Tables: The tables lack the references from which the data were taken. Please add appropriate citations to allow readers to trace the original sources.

3) Algorithm: It would be useful to make the proposed algorithm also in graphic form, to facilitate its understanding.

Agai, regarding the algorithm, it would be appropriate to clarify the following points:

- Share your direct experience with the procedure in question. How has your daily practice influenced the development of the algorithm?

- Provide some considerations on your case studies treated with this algorithm

- Highlight your skills and your role as experts in the field.

Providing this information would help validate the proposed algorithm and make it a more solid reference for the scientific community.

Author Response

We really appreciate your review.  Attached please find our response.  

Reviewer 2 Report

Comments and Suggestions for Authors

Very good, modern and valuable work. The extension of EBUS-TBNA goes in this direction, although for large infiltrating lesions a channel for the probe  can be created using a laser technique. However, the work requires corrections.

List od details to improve:

Section 2. (line 65) When using a 19-gauge needle 64 ot (to ?) perform the initial aspiration.......

Section 4. (line 120) Pneumothorax incidence of 0.03 (% ?) and bleeding incidence of 0.68%.... 

Section 6. (line 147) ....when compared to EBUS-TBNA Less (capital letter ?) common mediastinal...

Section 6. (page 5, line 180) ...alone was performed.27 (?) In addition...

Table 2. (line 223) NS= non-significant (NS = non......... ?)

Section 7. (line235) Other reported complications included cough (14%), Dyspnea  (capital letter ?) (1%), and hemoptysis (2%) [8,31].

Section 8. (line 256) For non-small lung cancer, however, Genetic  (capital letter?) testing was statistically...

Section 8. (line 262) ....was not statistically different. 59.6% Compared (capital letter?) to 66% with a p value.....

Comments on the Quality of English Language

Very good, modern and valuable work. The extension of EBUS-TBNA goes in this direction, although for large infiltrating lesions a channel for the probe  can be created using a laser technique. However, the work requires corrections.

List od details to improve:

Section 2. (line 65) When using a 19-gauge needle 64 ot (to ?) perform the initial aspiration.......

Section 4. (line 120) Pneumothorax incidence of 0.03 (% ?) and bleeding incidence of 0.68%.... 

Section 6. (line 147) ....when compared to EBUS-TBNA Less (capital letter ?) common mediastinal...

Section 6. (page 5, line 180) ...alone was performed.27 (?) In addition...

Table 2. (line 223) NS= non-significant (NS = non......... ?)

Section 7. (line235) Other reported complications included cough (14%), Dyspnea  (capital letter ?) (1%), and hemoptysis (2%) [8,31].

Section 8. (line 256) For non-small lung cancer, however, Genetic  (capital letter?) testing was statistically...

Section 8. (line 262) ....was not statistically different. 59.6% Compared (capital letter?) to 66% with a p value.....

Author Response

The authors sincerely appreciate the reviewer comments and their insight into the paper.  the following were corrected per the request,  

Section 2. (line 65) ot changed to to.  

Section 4. (line 120) Pneumothorax incidence of 0.03 % sign was added. 

Section 6. (line 147) ....when compared to EBUS-TBNA Less changed to less

Section 6. (page 5, line 180) ...alone was performed.27 changed to [27]

Table 2. (line 223) NS= non-significant (NS = non changed to not)

Section 7. (line235)  Dyspnea changed to dyspnea.  

Section 8. (line 256 Genetic changed to genetic

Section 8. (line 262) Compared changed to compared.

Round 2

Reviewer 1 Report

Comments and Suggestions for Authors

The Authors have provided sufficient clarifications on the requested points